# Experimental Study on Combustion Behavior of U-Shaped Cables with Different Bending Forms and Angles

**Changkun Chen \*, Wuhao Du and Tong Xu**

Institute of Disaster Prevention Science and Safety Technology, School of Civil Engineering,
Central South University, Changsha 410075, China; 224812212@csu.edu.cn (W.D.); tongxuxut@csu.edu.cn (T.X.)
* Correspondence: cckchen@csu.edu.cn; Tel.: +86-13187060583

**Abstract:** Cables are usually bent into a U-shape to cross obstacles during installation: this includes the upward-bending mode (UBM) and the downward-bending mode (DBM). An experimental study was conducted to investigate the combustion behavior of U-shaped cables with the above bending forms and different angles. The ignition point was set in the middle of the U-shaped cables and the temperature distribution, flame spread rate (FSR), mass loss rate (MLR), flame dimensional characteristics, etc. were measured and analyzed. The results showed that FSR and MLR are positively related to the bending angles, and the FSR is the highest in UBM 90°, close to 6.51 cm/min, which is four times higher than that in the bending angle 0° condition. In the UBM, the heat radiation and convection from the cable flame to the unburned region were more intense and the "eruptive fire phenomenon" occurred during the combustion process, leading to a sharp increase in the FSR in a short time. However, the thermal convection and radiation from the burning region to the unburned region were weakened in the DBM. Meanwhile, the molten outer sheath (PE) would flow along the cables, heating and igniting the unburned region in the DBM. In addition, the FSR, MLR, and peak temperature increased in the UBM compared to the DBM. The highest flame temperature occurred in UBD 90°, approximately 1023 °C.

**Keywords:** U-shaped cables; flame spread rate; bending angle; upward-bending mode (UBM); downward-bending mode (DBM)





## 1. Introduction

Cables, as carriers of electrical energy and signals, are widely used in fields such as electric power, communications, and industry. Fire accidents can easily be caused by faults such as damage to the insulation layer and the overload of cable conductors [1,2]. Cables are usually bent into a U-shape to cross obstacles in industry and construction. However, the stress in the bending section of the cable is usually relatively concentrated and the outer sheath of the aged cable is more prone to damage, leading to cable fire accidents and a greater hazard.

Currently, numerous scholars have conducted a series of experiments on the combustion characteristics and behavior of cable fire. Research institutions and fire departments in many countries have conducted research on the response and behavior in the event of a cable fire, such as FIPEC [3], CHRISTIFIRE [4], CAROLFIRE [5], etc., aiming to evaluate the performance under cable fire conditions and improve cable safety. Khan MM et al. [6] obtained a dimensional fire propagation index through experiments. Huang et al. [7] studied the influencing factors of the flame spread behavior of the cable and the study found that the flame spread rate increases with the increase of oxygen concentration. An et al. [8] conducted a fire experiment on PVC cables, demonstrating that, with the increase in the number of cables, both the length of the thermal decomposition combustion zone and the speed of flame propagation escalate. Zhao et al. [9] carried out an experimental study on the impact of the current on the spread of fire in polyethylene cables and the study found

that flames spreading downwards are more susceptible to the influence of the current. Zavaleta and Bascou [10–13] introduced an improved FLASH-CAT mode and analyzed the impact of cable fires on bridge racks under closed and mechanical ventilation conditions. Meinier et al. [14] determined the thermo-physical and combustion properties of the outer sheath using ethylene-vinyl acetate copolymer and polyethylene (PE) cables, and determined the rate of heat release from the materials, as well as the effective heat of combustion at different external heat flow densities. Xiao et al. [15] analyzed the combustion characteristics of PE by thermogravimetric experiments, such as the maximum heat release rate, the maximum smoke production rate, and so on. Basfar [16] characterized the polyethylene combustion properties in terms of the limiting oxygen index (LOI) and the average degree of combustion.

In addition, the tilt angle can change the shape of the flame and the rate of flame spread, which is a critical parameter affecting fire behavior. Hu et al. [17] investigated the behavior of cable flames at different tilt angles and found that the flame spread rate of the cable changes in a U-shape with the change of the angle. Chen et al. [18,19] studied the ignition behavior and temperature distribution of an inclined cable. Seung et al. [20,21] studied the fire spread characteristics of inclined cables. The study showed that the downstream flame spread rate and flame width increase with the increase of the tilt angle. Kobayashi [22,23] studied the phenomenon of melt during the combustion process of horizontal and vertical polyethylene cables and the flame propagation situation with different conductor materials. Lu et al. [24] conducted flame spread experiments on cables at different angles; the results showed that the flame spread rate first increases, and then decreases with the increase of horizontal wind speed.

In summary, the above work mainly focused on the spread of flame in straight cables and the effect of the angle of inclination on the combustion behavior. However, the bending section of U-shaped cables is usually subject to relatively concentrated stress, which, combined with the tendency of the aged cables' outer sheath, leads to them being more susceptible to damage and a serious fire hazard. Therefore, the behavior of U-shaped cables with different forms and angles was experimentally investigated in this work. The temperature distribution, in FSR and MLR et al.'s study, was recorded and analyzed. This work is expected to provide theoretical support for the fire protection design of U-shaped cables.

## 2. Experimental Set-Up

### 2.1. Experimental Apparatus and Measurement System

The combustion experiment equipment designed for U-shaped cables is shown in Figure 1. The three-core cross-linked polyethylene insulated cables (YJLY 3∗25) used in this experiment are composed of an outer sheath, filler, insulation layer, and aluminum core, from the outside to the inside. In the experiment, five U-shaped cables of 1.5 m in length were arranged in parallel with three types of bending angles: 30°, 60°, and 90°. The U-shaped cables were fastened to the cable bracket and the cable bracket was suspended on combustion experiment equipment by wire ties. A Sony high-speed camera and a FORTRIC thermal imager were arranged in front of the U-shaped cables, and used to record the combustion behavior and flame characteristics. A floor made of mica fireproof board was used to collect the high-temperature melt. Temperature-measuring points were placed on the floor at 10 cm intervals, to measure the temperature of the melt in real time. Two electric balances (with a precision of 0.1 g) were, respectively, positioned under the experiment equipment and floor, to monitor the total cable mass and melt mass. Thermocouples (with a precision of 0.1 °C) were connected to a multi-channel temperature collector (with an acquisition frequency of 1 Hz) to acquire the temperature. Six types of thermocouple brackets and cable brackets were manufactured for different U-shaped cable structures, which were used to fix the thermocouples and cables, as shown in Figure 2. The bending radius of the U-shaped cables was set to 15D (D is the cable diameter and 36 cm is fixed in this work) according to the standards. Each thermocouple bracket had three sets of

thermocouple trees (F, U, and D) and each thermocouple tree had nine measuring points. The F thermocouple was set at 5 cm above the cables to measure the flame temperature. The U and D thermocouple trees were, respectively, set on the top and bottom surfaces of the cables, to measure the surface temperature of the U-shaped cables.

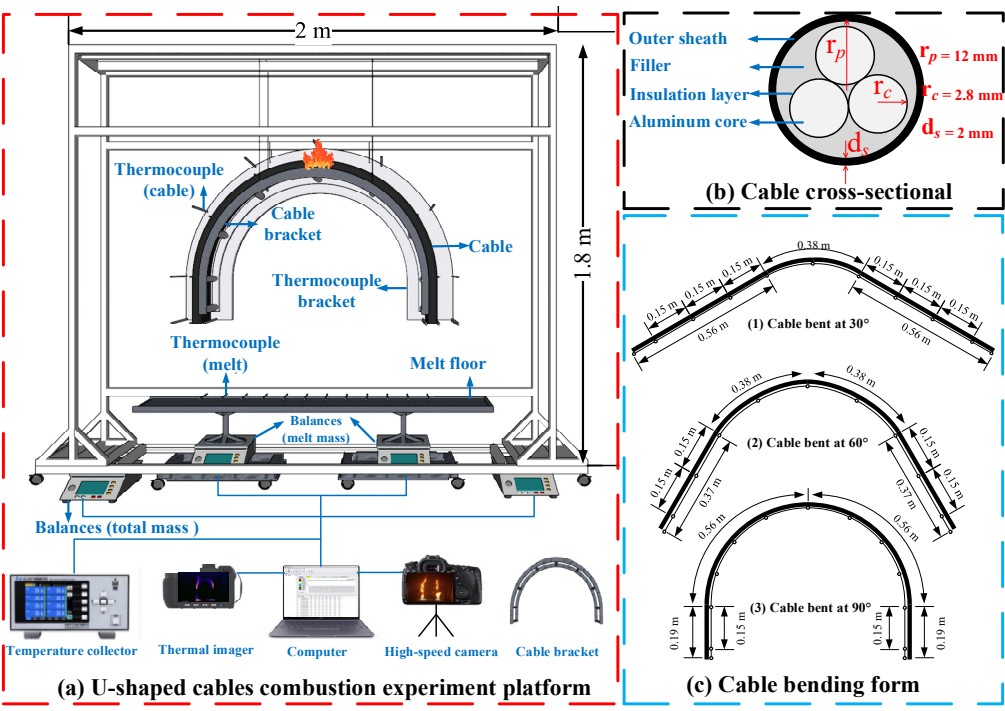

**Figure 1.** Schematic diagram of experiment equipment.

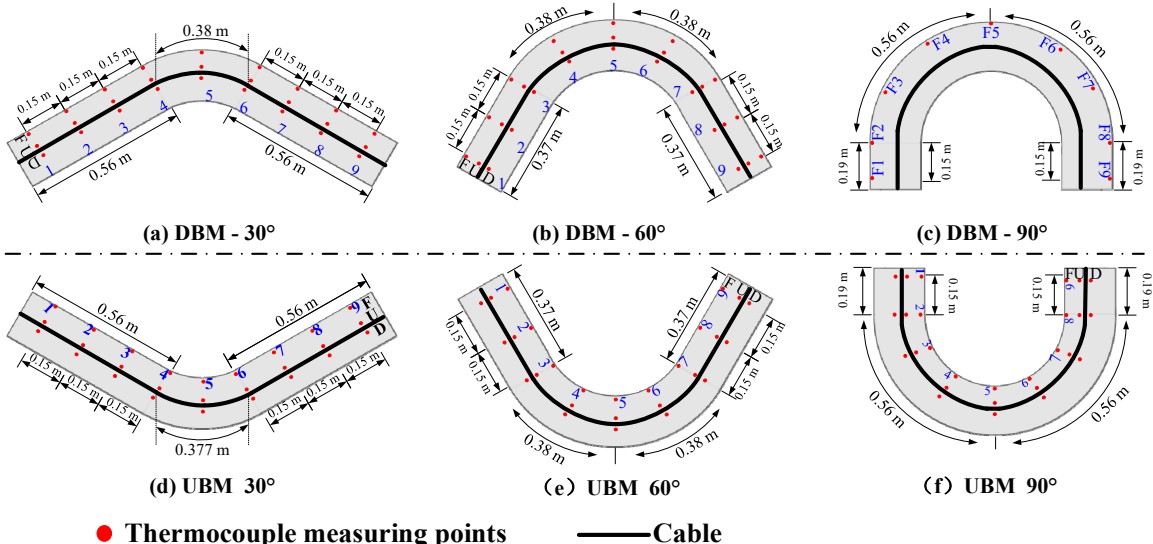

**Figure 2.** Schematic diagram of thermocouple bracket and measuring points.

Because of the stress concentration in the middle of the U-shaped cables, the outer sheath is susceptible to damage and has a higher fire risk. Therefore, the ignition point was set in the middle of the U-shaped cables. A butane torch was selected as the fire source and the heat release rate was about 2 kW, with an ignition time of 1 min. Table 1 gives the experimental conditions. Based on different bending forms and angles, seven groups of conditions were set, among which the bending angle of the C case (control group) is 0°, in the no-bending mode (NBM). For convenience of description, the upward-bending angle is denoted as a positive angle and the downward-bending angle as a negative angle in

the following description. Each set of experimental conditions was repeated at least three times. It should be noted that the experimental data naturally include a certain degree of random error, which will inevitably also be involved in the subsequent analysis and empirical model correlation. Hence, the current data errors need to be evaluated firstly. The uncertainty of the experimental data can be estimated by $X_i = X_i(\text{measured}) \pm \delta X_i$, $\sigma = \sqrt{\frac{\sum_{i=1}^{n}(X_i - \overline{X})^2}{n-1}}$, where the $\overline{X}$ represents the mean value of $n$ observations in multiple sample experiment; $\sigma$ is the standard deviation; and the $\delta X_i$ represents $2\sigma$ for a single sample test (i.e., $\delta X_i = 2\sigma$). The final estimation results imply that the maximum relative uncertainty of the temperature data is not more than 12.56% under the 95% confidence level. Similarly, the uncertainty of the mass loss data, flame spread rate, and flame parameters are evaluated to be no more than 9.28%, 11.06%, and 14.85%, respectively.

**Table 1.** Summary of experimental conditions.

| Test No. | Bending Form | Legend | Bending Angle ($\theta$) | Bending Radius | Ignition Location |
|---|---|---|---|---|---|
| A1 | Upward-bending mode (UBM) | | 30° | 15D (36 cm) | Middle |
| A2 | | | 60° | | |
| A3 | | | 90° | | |
| B1 | Downward-bending mode (DBM) | | 30° | | |
| B2 | | | 60° | | |
| B3 | | | 90° | | |
| C | No-bending mode (NBM) | | 0° | --- | |

### 2.2. Experimental Data Extraction

The flame dimensional parameters are crucial for analyzing the combustion behavior of U-shaped cables. The specific process is as follows (Figure 3):

(1) Use Premiere software to extract a video clip every 5 min and decompress the flame video into single flame images based on the time sequence;

(2) Employ the MATLAB-compiled program to transform the image into a gray-scale image, and then convert the picture into a binary image;

(3) Utilize the maximum between-class variance method for image segmentation to extract the flame shape [25];

(4) Perform a time-average process on the flame information at each pixel position to obtain the intermittent distribution contour of the flame. Define the length of the region with a probability of 0.5 as the characteristic width of flame ($W_f$). Define the length of the region with a probability of 0.5 on the upper surface of the cable and the region with a probability of 1 on the lower surface as the characteristic length of the pyrolysis region ($L_p$) [26].

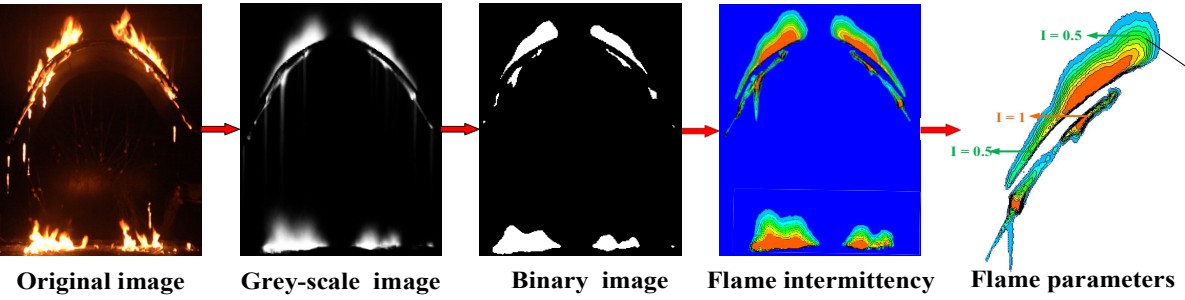

**Original image**    **Grey-scale image**    **Binary image**    **Flame intermittency**    **Flame parameters**

**Figure 3.** Schematic diagram of flame characteristic parameters.

## 3. Results and Discussion

### 3.1. Combustion Process

Figure 4 shows the front and thermal images of the combustion process of U-shaped cables with different bending forms and angles. It can be observed that the combustion process of U-shaped cables can be roughly divided into three phases: the bending section combustion stage, the inclined section combustion stage, and the melt combustion stage. The bending section combustion stage occurred in the initial stage of combustion, where the combustion area was limited and only the bending section showed a high temperature range in the thermal image. Furthermore, in the bending section, the flame plume ascended vertically and the effect of thermal convection and radiation on the unburned region was not significant. In the inclined section combustion stage, the fire spread rapidly, with the peak temperature exceeding 800 °C. In this stage, a large amount of outer sheath melt was burning on the floor and heating the end of the U-shaped cables. The flame on the cables' surface and the melt flame formed a combined coupled fire source, thus accelerating the combustion process. In the melt combustion stage, there were no more melt falling; only the melt on the floor was burning, and the fire gradually decreased until it was extinguished.

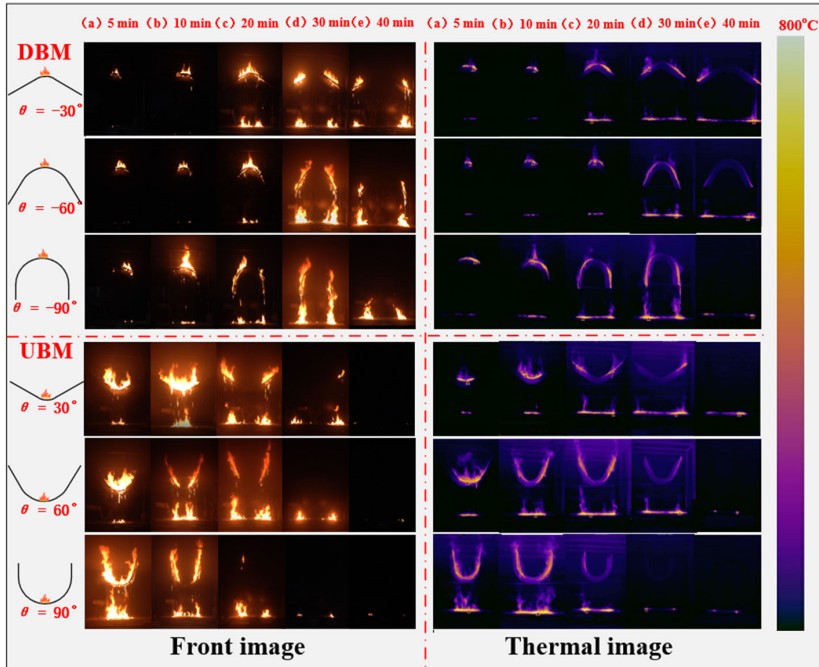

**Figure 4.** Combustion process of U-shaped cables with different bending forms and angles.

### 3.2. Flame Temperature Distribution

Figure 5 shows the flame temperature distribution during the combustion of U-shaped cables with different bending forms and angles. It can be found that the peak temperatures in the UBM are all above 750 °C, which is higher than those in the DBM. Furthermore, the time interval between temperature peaks is shorter in the UBM, while it is more dispersed in the DBM. This can be mainly attributed to two reasons: firstly, the combustion of the U-shaped cables in the UBM was mainly manifested as downstream flame propagation. The heat radiation and convection from the cable flame to the unburned region were more intense. Secondly, the cable melt, from the top end of the U-shaped cables, continued to flow down along the U-shaped cables and supply fuel to the burning region, during the combustion of the U-shaped cables in the UBM. It should be noted that a sudden increase in temperature is observed at approximately 35 min in UBM 90° (refer to Figure 5f). This is mainly due to the fusion of the flame at the end of the cables and the flame of the melt, leading to an increase in the heat release rate of the cables in a short time and a rapid rise in the flame temperature.

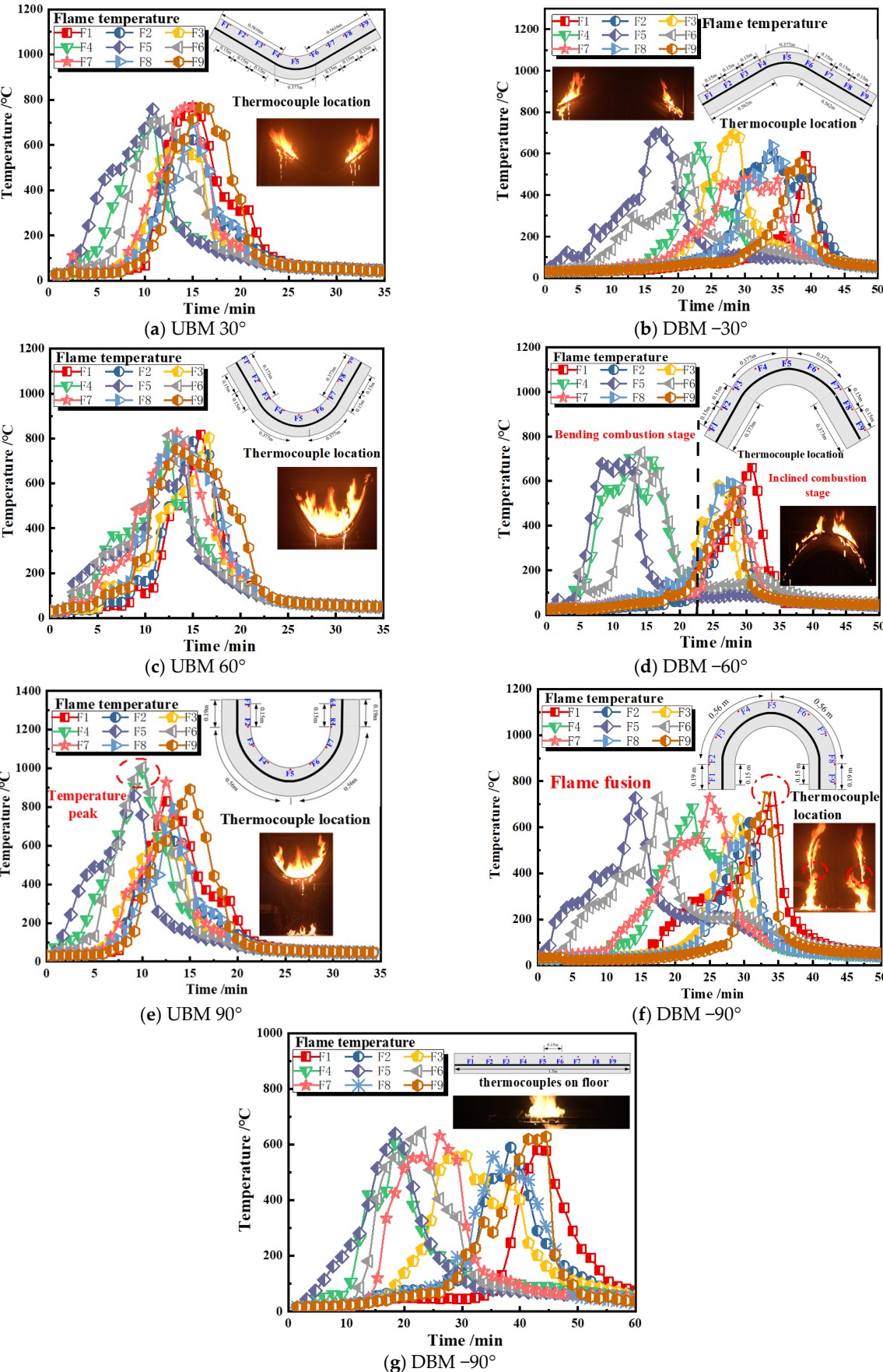

**Figure 5.** Flame temperature distribution with different bending forms and angles.

Figure 6 shows the peak flame temperatures with different bending forms and angles. It can be observed that the larger the bending angle of the U-shaped cables, the higher the peak flame temperature. The peak temperature of the U-shaped cables in the UBM is approximately 200 °C higher than that in the DBM. This is mainly due to the fact that, as the bending angle of the U-shaped increases, the distance between the cables on either side becomes shorter, increasing the thermal radiation and convection interactions between the flames at the ends of the cables. Additionally, it can be found that the peak temperature reaches approximately 1023 °C at the F4/F6 measuring point in UBD 90°, which is the maximum temperature under all conditions. This is mainly because the F4/F6 measuring point was located at the junction of the bending and inclined section. After the cables' top end was ignited, the melt would flow down the surface of the cables, increasing the heat received at this location. Meanwhile, the position was closer to the fire source on the other side of the cables; thus, it was subject to stronger thermal radiation.

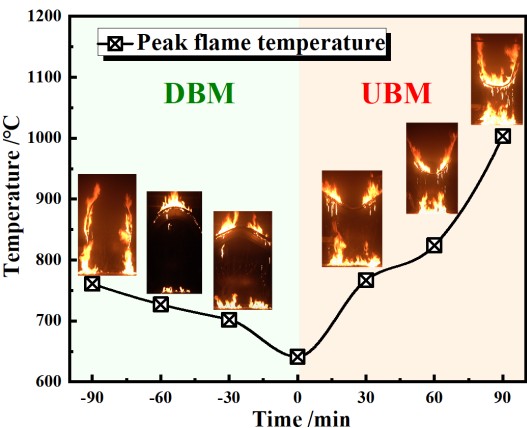

**Figure 6.** Peak flame temperatures with different bending forms and angles.

Figure 7 shows the temperature variation at the end of U-shaped cables (measuring point F9) with different bending forms and angles. It can be found that the smaller the bending angle, the lower the overall flame temperature at the cables' end, in which the lowest overall flame temperature (with a peak temperature of only 576 °C) occurs in the NBM. The flame temperature at the end of the U-shaped cables in the UBM is generally higher than that in the DBM in the same bending angle. Moreover, the FSR of the U-shaped cables is faster in the UBM. For example, it takes about 15 min to attain the peak temperature at the measuring point F9 in the UBM 90°, whereas it takes about 35 min in UBM 90°, with a difference of almost 20 min.

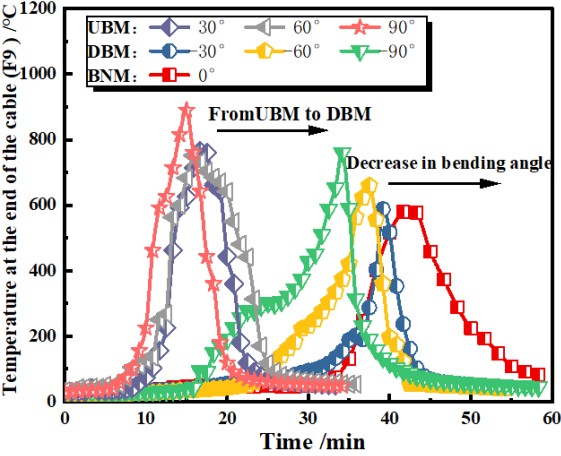

**Figure 7.** Temperature at the end of cables with different bending forms and angles.

Figure 8 shows the temperature distribution of the melt on the floor in DBM 30°. It can be found that the temperature (M6 and M7) in the middle of the floor begin to rise at about 7 min, with the dripping of the high-temperature melt. And the maximum temperature of the melt is about 550 °C. The overall temperature of the melt flame is approximately 500 °C, indicating that the melt poses a serious threat to the facilities below, requiring special attention in fire protection design and routine supervision. After approximately 43 min, the melt dripping from the cables below gradually decreased. Meanwhile, the temperature of the melt flame on the floor began to decrease until it approached the ambient temperature.

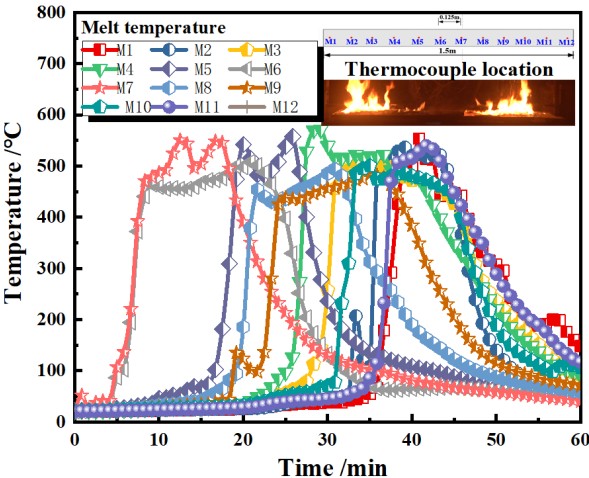

**Figure 8.** Temperature distribution of the melt (in UBD 30°).

### 3.3. Flame Spread Rate (FSR)

Generally, the ratio of the change in position of the flame front to time was defined as the flame spread rate (FSR). However, due to the rapid spread of flames at 90°, it is difficult to observe. Therefore, a new characteristic flame spread rate (FSR) was defined, and the definition method was considered as follows: as the outer sheath is mainly made of polyethylene, and the ignition point of polyethylene is 350 °C [17]. When the temperature at the measuring point reaches 350 °C, define the flame spreading to this measuring point, i.e., flame front location. Furthermore, the ratio of the distance between two measuring points to the time interval (when the temperature of the two measuring points reaches 350 °C), is defined as characteristic FSR in this work. Figure 9 shows the process of the variation of flame front location. It can be observed that the flame front moves the quickest in the UBM, followed by the DBM, with the lowest in the NBM.

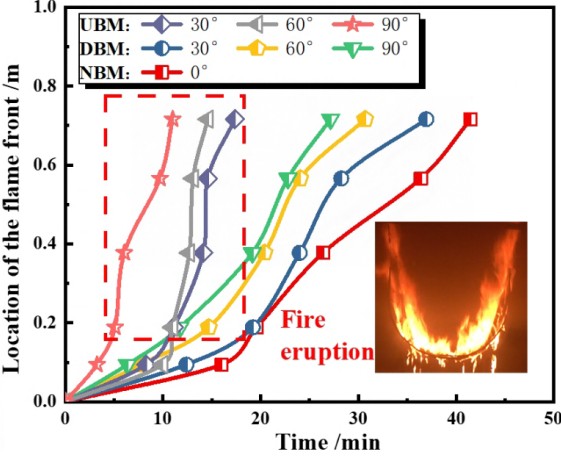

**Figure 9.** Variation in flame front location with time.

It should be noted that, during the combustion of the U-shaped cables in the UBM, the fire front spread dramatically in a short period of time. An eruptive fire phenomenon occurred, which implies a sharp increase in the FSR within a short period of time [27,28]. Combined with the heat transfer mode of flame spread in the U-shaped cables in Figure 10, the flame on the surface experienced limited air entrainment on one side, during the combustion of the U-shaped cables. And the differential pressure caused by the varying degrees of air entrainment on both sides led to flame adhesion on the cable surface, resulting in a flame attachment behavior. During the combustion of the U-shaped cables in the UBM, the attachment flame increased the contact surface between the flames and the cables, which improved the efficiency of mass and heat transfer. This effect led to large burning area, and, thus, to the eruptive fire phenomenon. Conversely, the preheating effect of attachment flame in DBM on the unburned outer sheath is weak; thus, no rapid increase in the FSR is observed.

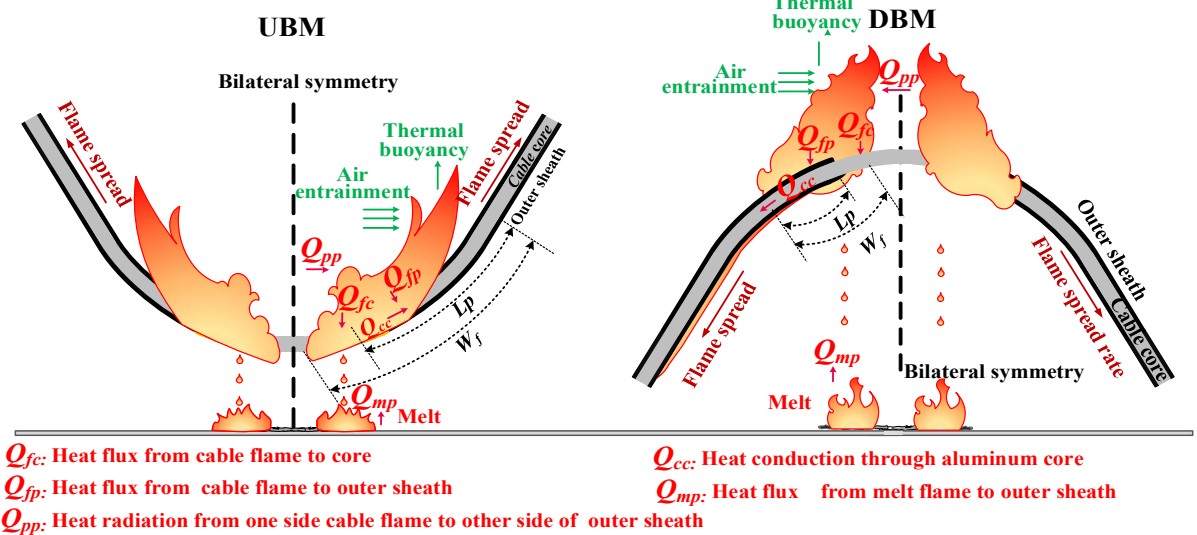

**Figure 10.** Heat transfer mode of flame spread in U-shaped cables.

The FSR was calculated by dividing the length of one side of the cables by the flame spread time. Figure 11 shows the diagram of the FSR with different bending forms and angles. For the bending form, the overall FSR in the UBM is approximately twice that in the DBM in the same bending angle. In addition to the differences of flame attachment, there are three reasons for this: (1) After the ignition in the middle of the U-shaped cables in the UBM, the overall propagation was mainly manifested as downstream flame propagation. During the propagation along the cables, the flame continued to rise and the flame length of the cable preheat area gradually increased; (2) After the U-shaped cables in the DBM were ignited, the melt flowed down the cables. The high-temperature melt attached to the surface of the cables was the main driving force for the FSR in the DBM; (3) For the U-shaped cables in the UBM, when the top end of the cables was ignited, the cable melt would flow down along the surface of the cables. After gathering at the bending end of the cables, the melt would fall to the floor. At this time, a large amount of melt is accumulated; thus, the heating effect of the unburned region of the cables was also more intense. Conversely, when the U-shaped cables in the DBM are heated, the melt would drip directly onto the floor along the cable surface, reducing and diminishing the heating effect on the unburned region of the cables.

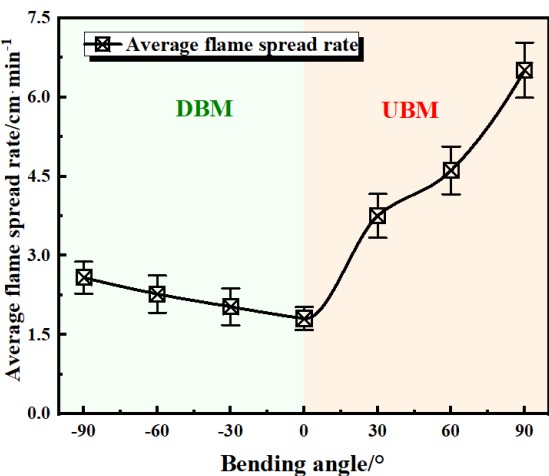

**Figure 11.** FSR with different bending forms and angles.

For the bending angle, it can be seen that the FSR increases with the bending angle. The FSR reaches the highest in UBD 90°, close to 6.51 cm/min, which is four times higher than that in the NBM. (It can be found that, in the DBM, $FSR_{-90°}$ = 2.58 cm/min, $FSR_{-60°}$ = 2.27 cm/min, and $FSR_{-30°}$ = 2.08 cm/min. In the UBM, $FSR_{30°}$ = 3.75 cm/min, $FSR_{60°}$ = 4.68 cm/min, and $FSR_{90°}$ = 6.51 cm/min. In the NBM, $FSR_{0°}$ = 1.80 cm/min). This is mainly due to the fact that the increasing bending angle intensifies the flame attachment effect, which increases two key parameters reflecting the FSR: the characteristic flame width ($W_f$) and characteristic length of the pyrolysis region ($L_p$). Figure 12 shows the average $W_f$ and $L_p$ with different bending forms and angles. It can be observed that both $W_f$ and $L_p$ increase as the bending angle increases, which shows a strong positive correlation with the FSR of the cables. The main reasons are as follows: (1) The longer the $W_f$ and $L_p$, the larger the area of the cables involved in combustion. The increase in the combustion surface area and the specific surface area intensifies the heat exchange process between the flame and the cable surface, resulting in more melting of the cables' outer sheath and the generation of flammable gases; (2) The longer the $L_p$ and $W_f$, the greater the flame air contact area, which further enhances the air convection effect, increasing the supply of oxygen.

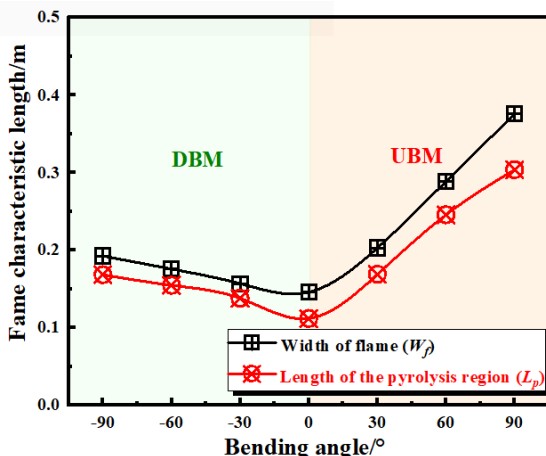

**Figure 12.** Flame characteristic length with different bending forms and angles.

### 3.4. Total Mass and Melt Mass

Figure 13 shows the variation in the total mass and the melt mass during the combustion process of U-shaped cables with different bending forms and angles. The maximum mass loss during cable combustion ranges from 2.47 kg to 1.91 kg, and the maximum cumulative melt mass ranges from 1.56 kg to 0.97 kg. The mass curve can also be divided into three typical phases: the bending section combustion stage (stage I), the inclined

section combustion stage (stage II), and the melt combustion stage (stage III). During the bending section combustion stage, the total mass loss decreases gradually, while the melt mass increases slowly. This is mainly because the overall FSR was low in the bending section, with only a small portion of the outer sheath being thermally decomposed and burned. As the cable fire spread to the inclined section, the total mass loss increased rapidly and the melt mass also accumulated rapidly. Due to the rapid increase in $W_f$ and $L_p$ in the inclined section, a large area of thermal decomposition burning occurred, leading to a rapid increase in the total mass loss. A large amount of melt fell, accompanied by a large amount of melt being quickly burned and consumed. In the melt combustion stage, the combustibles on the surface of the cables were essentially exhausted; there was no more melt falling. Only the melt was burning on the floor, leaving the cables with almost only the exposed aluminum core. The melt mass on the floor gradually decreased until the flame was extinguished.

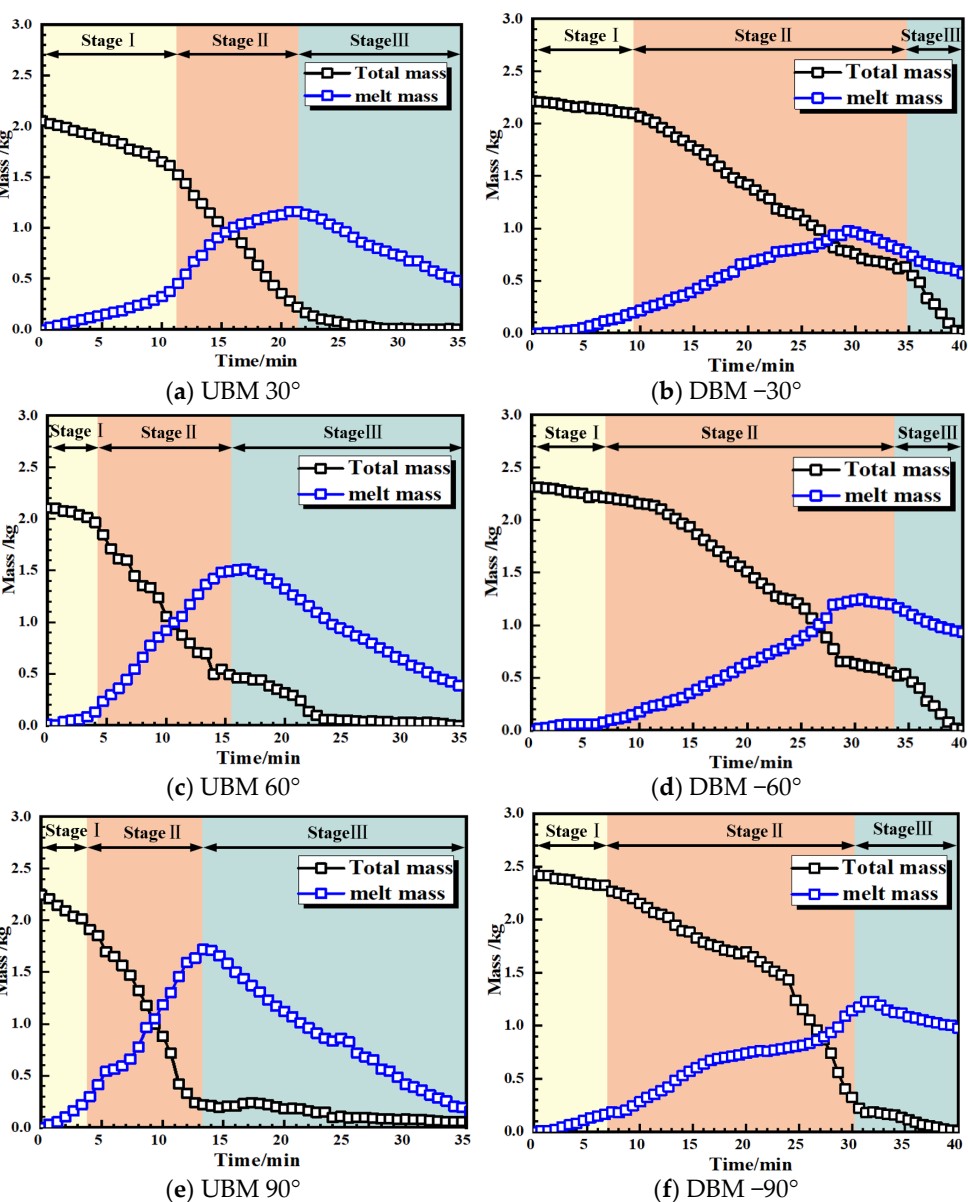

**Figure 13.** Variation in total mass and melt mass (stage I: bending combustion section stage, stage II: inclined section combustion stage, and stage III: melt combustion stage).

Figure 14 shows the total mass loss and the mass loss rate (MLR, total mass loss divided by burning time) of U-shaped cables with different bending forms and angles. (It

can be found that, in the DBM, $MLR_{-90°} = 0.102$ kg/min, $MLR_{-60°} = 0.076$ kg/min, and $MLR_{-30°} = 0.062$ kg/min. In the UBM, $MLR_{30°} = 0.101$ kg/min, $MLR_{60°} = 0.152$ kg/min, and $MLR_{90°} = 0.206$ kg/min. In the NBM, $MLR_{0°} = 0.043$ kg/min) The variation trend of total mass loss and mass loss rate is basically the same for bending angle, the larger the bending angle of U-shaped cables, the greater the total mass loss and mass loss rate. However, it should be noted that the combustion of U-shaped cables in DBM is more complete than that in UBM, which means larger total mass loss and less residue in DBM. The main reason is that the flame enveloped the cables for a longer period of time, resulting in more cables mass being consumed during combustion. The burnt morphology of a single cable is shown in Figure 15. Due to the extremely high combustion temperature, the cable core fractured under the combined effect of stress and high temperature in UBD 90°.

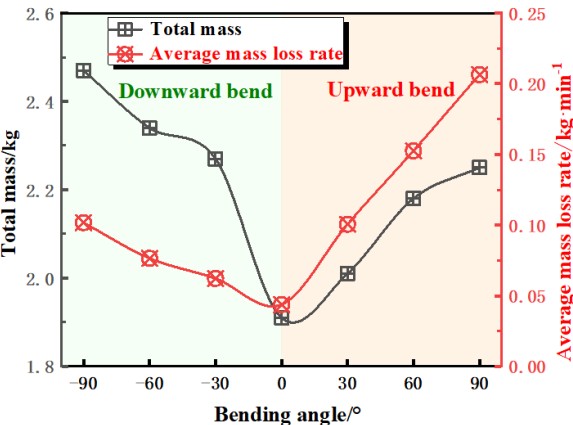

**Figure 14.** Total mass loss and mass loss rate with different bending forms and angles.

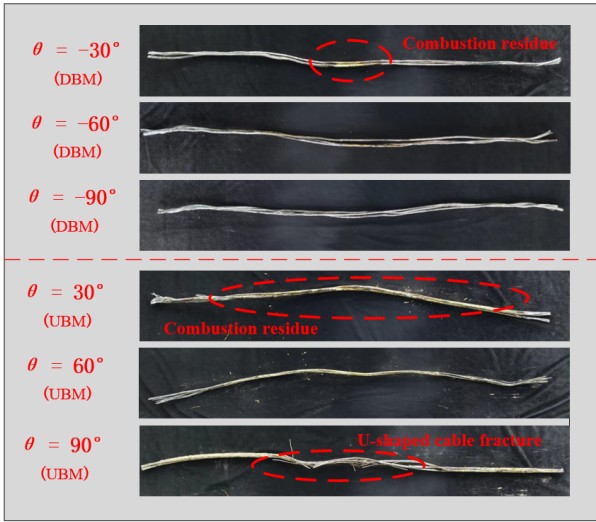

**Figure 15.** Burnt morphology of a single cable.

## 4. Conclusions

In this work, the combustion behavior of U-shaped cables with different forms and angles was experimentally investigated, and the temperature distribution, flame spread rate (FSR), mass loss rate (MLR), flame dimensional characteristics, etc. were measured and analyzed. The main conclusions are as follows:

(1)    When the middle part of the U-shaped cables was ignited, the burning behavior of U-shaped cables with different bending forms and bending angles varied considerably. And the combustion could be divided into three typical phases: the bending section combustion stage, the inclined section combustion stage, and the melt com-

bustion stage. A large amount of melt dripped onto the floor, with the temperature approaching 500 °C, indicating a high fire hazard.

(2)　For the same angle, the FSR is highest in the UBM, about 6.51 cm/min, which is approximately twice as high as in the DBM, and four times higher than that in the NBM. This is mainly because the flame height continued to increase during the flame propagation, increasing the preheat length in the UBM (i.e., the downstream flame). Additionally, the flame attachment behavior improved the efficiency of the mass and heat transfer, thereby accelerating the FSR.

(3)　The U-shaped cables in the UBM had a higher flame temperature. The peak temperature of the U-shaped cables in the UBM is approximately 200 °C higher than that in the DBM. As the bending angle increases, the time to reach the temperature peak decreases and the flame temperature increases. The highest flame temperature occurred in UBM 90°, which was approximately 1023 °C.

(4)　The larger the bending angle of the U-shaped cables, the higher the mass loss rate. The maximum mass loss was 0.2 kg/min. The flame enveloped the cables for a longer time, resulting in more cable mass being consumed and less residue in the DBM. The U-shaped cables in DBM 90° had the highest total mass loss of almost 2.47 kg.

Finally, it should be pointed out that the current experimental study investigated the effect of different forms and angles on the combustion behavior of U-shaped cables under middle ignition. However, there are significant differences in the combustion behavior of cables at different ignition positions. These issues should be explored and revealed in future work.

**Author Contributions:** C.C.: conceptualization, methodology, supervision, project administration, and writing—review and editing. W.D.: writing—original draft, data curation, and investigation. T.X.: validation, visualization, and editing. All authors have read and agreed to the published version of the manuscript.

**Funding:** This work was supported by the National Natural Science Foundation of China (Grant No. 72091512). The authors appreciate the supports deeply.

**Institutional Review Board Statement:** Not applicable.

**Informed Consent Statement:** Not applicable.

**Data Availability Statement:** The raw data are held by the author and may be made available upon request.

**Conflicts of Interest:** The authors declare no conflict of interest.

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
