# Peer review of "Experimental Study on Combustion Behavior of U-Shaped Cables with Different Bending Forms and Angles"

_fire, doi:10.3390/fire6090348_

Round 1
Reviewer 1 Report
In this paper, an experimental study was conducted to investigate the combustion behaviour of U-shaped cables with the above bending forms and different angles. The main comments are as follows:
As an experimental research, the reliability of the experiment is very important, so for the experimental research in this paper, what is the result of the experimental error analysis? Is there a replication study of the experiment conducted?
The English in the paper needs to be carefully checked.
The English in the paper needs to be carefully checked.
Author Response
Primary response:
Sincerely thanks for your consideration of giving us the opportunity to revise and resubmit our manuscript. We sincerely express our gratitude to editor and reviewers for the valuable comments. The editor’s and reviewers’ comments are laid out below in black text and specific concerns have been numbered. The revision has been finished, and the changes, additions and replies to reviewers' comments are listed below. All the changes, including text, equations and figures (captions), are also highlighted point by point using red text in the revised manuscript.Symbols and pictures are not shown in full in the website, please view the full content in Author's Notes File.
Once again, we really appreciate the works of editors and reviewers. These comments and suggestions have greatly helped to improve the manuscript quality. Please let us know if further modifications are required. Your reply will be highly appreciated. Sincerely thanks.
-------------------------------------------------------------------------------------------------------
-------------------------------------------------------------------------------------------------------
Comments from Reviewer #1:
-------------------------------------------------------------------------------------------------------
² Comment 1: As an experimental research, the reliability of the experiment is very important, so for the experimental research in this paper, what is the result of the experimental error analysis? Is there a replication study of the experiment conducted?
-
- Response: Sincerely thanks to the reviewer’s timely reminder. Each set of experimental conditions was repeated at least three times. It should be pointed out that the experimental data naturally includes a certain degree of random error, which will inevitably be also involved in the subsequent analysis and empirical model correlation. Hence, the current data errors need to be evaluated firstly. In order to ensure the accuracy of the critical velocity in this study, we use the Eq. (1) to estimate the experimental data error.
,
(1)where the Xi(measured) represents the observation in a single sample experiment or the mean of a series of n observations in a multiple sample experiment; σ is the standard deviation of measurements. The δXi represents 2σ for a single sample test (i.e. δXi=2σ ). The final estimation results imply that the maximum relative uncertainty of temperature data is not more than 12.56 % under the 95% confidence. Similarly, the uncertainty of mass loss data, flame spread rate and fame parameters are evaluated to be no more than 9.28%, 11.06%, and 14.85% respectively.
Once again, sincerely thanks to reviewer for the constructive comments. We have also added corresponding explanations in the revised manuscript. Thanks again, sincerely.
The specific changes/additions to manuscript are outlined below in red text:
It should be noted that the experimental data naturally includes a certain degree of random error, which will inevitably be also involved in the subsequent analysis and empirical model correlation. Hence, the current data errors need to be evaluated firstly. The uncertainty of experimental data can be estimated by
, . Where the represents the mean value of n observations in multiple sample experiment; σ is the standard deviation. The δXi represents 2σ for a single sample test (i.e. δXi=2σ). The final estimation results imply that the maximum relative uncertainty of temperature data is not more than 12.56 % under the 95% confidence. Similarly, the uncertainty of mass loss data, flame spread rate and fame parameters are evaluated to be no more than 9.28%, 11.06%, and 14.85% respectively.
------------------------------------------------------------------------------------------------------
² Comment 2: The English in the paper needs to be carefully checked.
- Response: Sincerely thanks to the reviewer’s timely reminder. Words in the paper have been carefully checked for spelling and grammar.
The specific changes/additions to manuscript are outlined below in red text:
(Page 10, line 273-275 ,of revised manuscript)
For the bending angle, it can be seen that the FSR increases with the bending angle. The FSR reaches the highest in UBD 90°, close to 6.81cm/min, which is four times high-er than that in NBM.
(Page 10, line 278-280 ,of revised manuscript)
Conversely, when U-shaped cables in DBM are heated, the melt would drip directly onto the floor along the cables surface, reducing diminishes the heating effect on the unburned region of the cables.
(Page 14, line 369-371 ,of revised manuscript)
Finally, it should be pointed that the current experimental study investigated the effect of different forms and angles on the combustion behaviour of U-shaped cables under middle ignition.
-------------------------------------------------------------------------------------------------------
Once again, we deeply appreciate the reviewers' recognition and precious comments, as well as the editor's patient response. These constructive suggestions are very important for our current work. Sincerely hope that with reviewers' and editor's guidance, we could improve the quality of our manuscript to the greatest extent and finally reach the published level in Fire.
Once again, sincerely thanks.

Reviewer 2 Report
The manuscript reports the results of experimental studies of the combustion behavior of U-shaped cables with different angles. The manuscript fits the scope of MDPI Fire but needs major revision. My detailed comments are given below.
(1) The literature review is incomplete. Since the manuscript deals with combustion of polyethylene cables, the authors must provide the fundamental information on the combustion parameters of polyethylene sheets, rods, balls, etc. with the corresponding references.
(2) The experimental technique is presented inadequately. First, the authors must clearly indicate how they measure the flame parameters (solely by thermocouples, or by both thermocouples and FORTRIC thermal images). Second, the authors must show the primary measurement data, like records of thermocouples, to see the response time, the intermittency, etc. Third, the authors must show the results of thermocouple/image calibration using a source with the known temperature. Fourth, the authors must specify the errors of all measured parameters. The origin of error bars in Figure 11 is not clear to me.
(3) The way the authors "measure" the flame temperature and its distribution, as well as the flame spread rate looks dubious. Since the flame is unsteady, it is not that easy to measure its temperature by a thermocouple. The standard ignition temperature of polyethylene (~350 C) cannot be directly used for the experiments under consideration because the standard conditions for the measurements of the ignition temperature are completely different from those realized in the authors' experiments. The authors must thoroughly justify their approach.
(4) It is not clear what kind of new information is provided by Figure 12. Keeping in mind the unsteady (probabilistic) nature of the flame and large accompanying uncertainties I could not distinguish between the curves for the flame width and the pyrolysis zone length in this figure, as they nearly merge.
(5) In view of comments (1)-(4), the conclusions made by the authors look weakly substantiated.
Minor editing of English language is required
Author Response
Primary response:
Sincerely thanks for your consideration of giving us the opportunity to revise and resubmit our manuscript. We sincerely express our gratitude to editor and reviewers for the valuable comments. The editor’s and reviewers’ comments are laid out below in black text and specific concerns have been numbered. The revision has been finished, and the changes, additions and replies to reviewers' comments are listed below. All the changes, including text, equations and figures (captions), are also highlighted point by point using red text in the revised manuscript.Symbols and pictures are not shown in full in the website, please view the full content in Author's Notes File.
Once again, we really appreciate the works of editors and reviewers. These comments and suggestions have greatly helped to improve the manuscript quality. Please let us know if further modifications are required. Your reply will be highly appreciated. Sincerely thanks.
-------------------------------------------------------------------------------------------------------
-------------------------------------------------------------------------------------------------------
Comments from Reviewer #2:
-------------------------------------------------------------------------------------------------------
² Comment 1: The literature review is incomplete. Since the manuscript deals with combustion of polyethylene cables, the authors must provide the fundamental information on the combustion parameters of polyethylene sheets, rods, balls, etc. with the corresponding references.
- Response: Sincerely thanks to the reviewer’s timely reminder. Literature on the fundamental information on the combustion parameters of polyethylene has been added to the revised manuscript.
The specific changes/additions to manuscript are outlined below in red text:
(Page 2, line 49-56 of revised manuscript)
Meinier et al. [14] determined the thermo-physical and combustion properties of the outer sheath using ethylene-vinyl acetate copolymer and polyethylene (PE) cables, and determined the rate of heat release from the materials as well as the effective heat of combustion at different external heat flow densities. Xiao et al. [15] analyzed the combustion characteristics of PE by thermogravimetric experiments, such as the maximum heat release rate, the maximum smoke production rate and so on. Basfar [16] characterized polyethylene combustion properties in terms of limiting oxygen index (LOI) and average degree of combustion.
[14] Meinier, R.; Sonnier, R.; Zavaleta, P.; Suard, S.; Ferry, L. Fire behavior of halogen-free flame retardant electrical cables with the cone calorimeter. J. Hazard. Mater. 2018, 342 ,306-316
[15] Xiao, M.; Liang, D.; Shen,H. Research on Flame Retardancy and Combustion Characteristics of PE and PE-MH-NC Cable Materials. Procedia Engineering. 2016, 135, 243-247.
[16] Basfar ,A.A. Flame retardancy of radiation cross-linked poly(vinyl chloride) (PVC) used as an insulating material for wire and cable. Polym. Degrad. Stabil. 2002, 77, 221-226.
-------------------------------------------------------------------------------------------------------
² Comment 2: The experimental technique is presented inadequately. First, the authors must clearly indicate how they measure the flame parameters (solely by thermocouples, or by both thermocouples and FORTRIC thermal images). Second, the authors must show the primary measurement data, like records of thermocouples, to see the response time, the intermittency, etc. Third, the authors must show the results of thermocouple/image calibration using a source with the known temperature. Fourth, the authors must specify the errors of all measured parameters. The origin of error bars in Figure 11 is not clear to me.
- Response: Sincerely thanks to reviewer for these precious comments. One-by-one responses to the above comments are as follows:
(1) Flame parameter was measured by high-speed camera in this work. Flame parameter handling is mentioned in the manuscript. The specific process is as follows (Figure 1):
1) Use Premiere software to extract a video clip shot by high-speed camera every 5 minutes and decompress the flame video into single flame images based on the time sequence;
2) Employ the MATLAB compiled program to transform the image into a gray-scale image and then convert the picture into a binary image;
3) Utilize the maximum between-class variance method for image segmentation to extract the flame shape;
4) Perform a time average process on the flame information at each pixel position to obtain the intermittent distribution contour of the flame. Define the length of the region with a probability of 0.5 as the characteristic width of flame (Wf). Define the length of the region with a probability of 0.5 on the upper surface of the cable and the region with a probability of 1 on the lower surface as the characteristic length of the pyrolysis region (Lp).

Figure 1. Schematic diagram of flame characteristic parameters.
(2) The thermocouple collector used in the experiment collects at a frequency of 1Hz, i.e., six temperature data are collected in one second. The figure below gives the 1 and processed image of the thermocouple collector collecting a total of 3000 temperature data in 50 minutes (right figure). For presentation purposes, 49 of these temperature data are ignored in the plotting process (right figure).

Figure 2. Flame temperature distribution in UBM 30°.
(3) We have used a source with the known temperature for thermocouple/image calibration before starting the experiment. The calibration process is as follows:
As described before [Brohez S, Delvosalle C, Marlair G. A two-thermocouple probe for radiation corrections of measured temperatures in compartment fires [J]. Fire Safety Journal, 2004, 39(5):399-411.], the temperature measured by thermocouple may be different from the actual gas temperature, and its error mainly has the following several sources: catalytic heating reactions near thermocouple probes, thermal inertia of thermocouples, heat conduction along wires, and radiant heating or cooling of junctions [M. Heitor, A. Moreira. 1993. Thermocouples and sample probes for combustion studies [J]. Progress in energy and combustion science, 19: 259 -278.]. At present, the method of temperature correction using thermocouples with different diameters has been established, which is based on the basic heat transfer theory. The expression is as follows:
|
(1) |
Where m, c and s are the mass, specific heat by volume and surface area of thermocouple beads, respectively. T is the temperature of the thermocouple bead, t is time, h is the convective heat transfer coefficient of hot gas and thermocouple beads, ?? is hot gas temperature, ?0 is effective environmental temperature, ? is emittance of thermocouple beads and σ is Stepan-Boltzmann constant.
If the approximate steady state temperature is measured by the bare thermocouple beads placed in the flame, and flame temperature pulsation has little effect on transient heating term, then the Eq.1 can be rewritten as:
|
|
(2) |
Here, the convective heat transfer coefficient can be estimated by Nu=hd/kg, d is the diameter of the bead, ?g is thermal conductivity for the fires. Nu can be calculated using the relation proposed by Zukauskas:
|
|
(3) |
Where Re= ud/vg. u and vg are the velocity and kinetic viscosity of the plume flow direction, respectively. Pr is Prandtl number.
Based on Eq.2 and Eq.3, a temperature correction method using two thermocouples with different diameters is proposed. The relationship is shown as follows:
|
|
(4) |
Where the following table s and b represent thermocouples with smaller bead diameter and larger bead diameter respectively.
In this study, another k-type thermocouple with a bead diameter of 0.8mm was used to correct the temperature value in the experiment. Combined with Eq.1 and Eq.4, the calibrated temperature (Tg) value can be finally calculated, and the calibrated process is shown below:

Figure 3. Temperature calibration results.
(4) In order to ensure the accuracy of the critical velocity in this study, we use the Eq. (1) to estimate the experimental data error and error bars.
,
(1)
whereXi the (measured) represents the observation in a single sample experiment or the mean of a series of n observations in a multiple sample experiment; σ is the standard deviation of measurements. The δ Xi represents 2σ for a single sample test (i.e. δXi=2σ). The final estimation results imply that the maximum relative uncertainty of temperature data is not more than 12.56 % under the 95% confidence. Similarly, the uncertainty of mass loss data, flame spread rate and fame parameters are evaluated to be no more than 9.28%, 11.06%, and 14.85% respectively. Error bars in Figure 11 is the standard deviation of measurements, which is calculated Eq. (1).
The specific changes/additions to manuscript are outlined below in red text:
(Page 4, line 128-144 of revised manuscript)
Flame dimensional parameters are crucial for analyzing the combustion behaviour of U-shaped cables. The specific process is as follows (Figure 3):
1) Use Premiere software to extract a video clip every 5 minutes and decompress the flame video into single flame images based on the time sequence;
2) Employ the MATLAB compiled program to transform the image into a gray-scale image and then convert the picture into a binary image;
3) Utilize the maximum between-class variance method for image segmentation to extract the flame shape [25];
4) Perform a time average process on the flame information at each pixel position to obtain the intermittent distribution contour of the flame. Define the length of the region with a probability of 0.5 as the characteristic width of flame (Wf). Define the length of the region with a probability of 0.5 on the upper surface of the cable and the region with a probability of 1 on the lower surface as the characteristic length of the pyrolysis region (Lp) [26].

Figure 3. Schematic diagram of flame characteristic parameters.
(Page 2, line 89-93 of revised manuscript)
Two electric balances (with a precision of 0.1 g) were respectively positioned under the experiment equipment and floor, to monitor the total cable mass and melt mass. Thermocouples (with a precision of 0.1 ℃) were connected to a multi-channel temperature collector (with a acquisition frequency of 1 Hz) to collect temperature.
(Page 3, line 114-124 of revised manuscript)
It should be noted that the experimental data naturally includes a certain degree of random error, which will inevitably be also involved in the subsequent analysis and empirical model correlation. Hence, the current data errors need to be evaluated firstly. The uncertainty of experimental data can be estimated by
. Where the represents the mean value of n observations in multiple sample experiment; is the standard deviation. The δ Xi represents 2σ for a single sample test (i.e. δ Xi=2σ ). The final estimation results imply that the maximum relative uncertainty of temperature data is not more than 12.56 % under the 95% confidence. Similarly, the uncertainty of mass loss data, flame spread rate and fame parameters are evaluated to be no more than 9.28%, 11.06%, and 14.85% respectively.
-------------------------------------------------------------------------------------------------------
² Comment 3: The way the authors "measure" the flame temperature and its distribution, as well as the flame spread rate looks dubious. Since the flame is unsteady, it is not that easy to measure its temperature by a thermocouple. The standard ignition temperature of polyethylene (~350 C) cannot be directly used for the experiments under consideration because the standard conditions for the measurements of the ignition temperature are completely different from those realized in the authors' experiments. The authors must thoroughly justify their approach.
- Response: Sincerely thanks to reviewer for these precious comments. One-by-one responses to the above comments are as follows:
(1) Firstly, we deeply appreciate your precious and pertinent comments. In the cable combustion process the flame is continuously pulsating, the use of thermocouples to collect the flame temperature is not easy. Secondly, Thermocouple acquisition frequency is once per second, in one second the flame in a certain area many times pulsation, hot spot thermocouple collector in one second the average temperature characterization of the flame temperature. Finally, Flame pulsation is a normal flame phenomenon in the combustion process, which is an unavoidable problem in the process of measuring flame temperature. Most of the previous studies use thermocouples to measure the flame temperature, and this method can reflect the flame law to a certain extent.
(2) Firstly, we very much agree with you about the standard ignition temperature of polythene. Standard ignition temperatures are usually measured under specific conditions that differ from our experimental conditions. Secondly, In the process of processing the data, we used the video image method to process the fire spread rate. The main steps in the video image method processing fire spread rate are: (1) Use Premiere software to extract a video clip shot by high-speed camera every 5 minutes and decompress the flame video into single flame images based on the time sequence; (2) Employ the MATLAB compiled program to transform the image into a gray-scale image and then convert the picture into a binary image; (3) Utilize the maximum between-class variance method for image segmentation to extract the flame shape; (4)Perform a time average process on the flame information at each pixel position to obtain the intermittent distribution contour of the flame.(5) Define the front of the region with a probability of 0.5 as the flame region, and record the flame region front position versus time curve. We compared the fire spread rate determined by the video image method with the fire spread rate determined by the polyethylene ignition method, and the overall pattern of the two is the same, which proves the feasibility of the method to a certain extent. Finally, we emphasize in the paper that the fire spread rate calculated using the ignition point of polyethylene is a characteristic fire spread rate, which does not reflect the true fire spread rate but is used to reflect the trend of the fire spread rate of U-shaped cables with different bending forms and angles.
Based on the reviewer’s comments, we have also added corresponding.

Figure 4. Flame spread rate (FSR)under two methods
The specific changes/additions to manuscript are outlined below in red text:
(Page 8, line 235-238 of revised manuscript)
Since the experimental conditions deviated from the standard conditions, the FSR calculated using the ignition point of polyethylene is a characteristic fire spread rate, which does not reflect the true fire spread rate but is used to reflect the trend of the fire spread rate of U-shaped cables with different bending forms and angles.
-------------------------------------------------------------------------------------------------------
² Comment 4: It is not clear what kind of new information is provided by Figure 12. Keeping in mind the unsteady (probabilistic) nature of the flame and large accompanying uncertainties I could not distinguish between the curves for the flame width and the pyrolysis zone length in this figure, as they nearly merge.
- Response: Figure 12 shows the characteristic characteristic flame width and characteristic length of the pyrolysis region, which corresponds to the flame width and pyrolysis length for the case of the highest flame height, and is used to quantitatively analyze the burning speed at different bending angles.

Figure 5. Fame characteristic length with different bending forms and angles.
-------------------------------------------------------------------------------------------------------
² Comment 5: In view of comments (1)-(4), the conclusions made by the authors look weakly substantiated.
- Response: Sincerely thanks to reviewer for these precious comments. We have made changes to the conclusions, including the inclusion of some qualitative findings to increase the credibility of the conclusions. Based on the reviewer’s comments, we have also added corresponding.
The specific changes/additions to manuscript are outlined below in red text:
(Page 14, line 348-370 of revised manuscript)
1) When the middle part of U-shaped cables was ignited, the burning behaviour of U-shaped cables with different bending forms and bending angles varied considerably. And the combustion could be divided into three typical phases: the bending section combustion stage, the inclined section combustion stage and the melt combustion stage. A large amount of melt dripped onto the floor, with the temperature approaching 500℃, indicating a high fire hazard.
2) For the same angle, the FSR is highest in UBM, about 6.81 cm/min, which is approximately twice as high as in DBM, four times higher than that in NBM. This is mainly because the flame height continued to rise during the flame propagation, increasing the preheat length in UBM (i.e., the downstream flame). Additionally, the flame attachment behaviour improved the efficiency of mass and heat transfer, thereby accelerating the FSR.
3) The U-shaped cables in the UBM had a higher flame temperature and more centralized temperature distribution profile. The peak temperature of U-shaped cables in UBM is approximately 200 ℃ higher than that in DBM. As the bending angle increases, the time to reach the temperature peak at each measuring point decreases and the flame temperature increases. The highest flame temperature occurred in UBM 90°, which was approximately 1023 ℃.
-------------------------------------------------------------------------------------------------------
Once again, we deeply appreciate the reviewers' recognition and precious comments, as well as the editor's patient response. These constructive suggestions are very important for our current work. Sincerely hope that with reviewers' and editor's guidance, we could improve the quality of our manuscript to the greatest extent and finally reach the published level in Fire.
Once again, sincerely thanks.

Reviewer 3 Report
This work presented the tested results and analysis on bending and angle effect on combustion behavior of U-shaped cables. Overall, this work is academically sound. A minor issue to address would be as this is an experimental study, please include uncertainties for the measured and deduced parameters.
Minor language issues such as a few misspellings in the manuscript should be fixed.
Author Response
Primary response:
Sincerely thanks for your consideration of giving us the opportunity to revise and resubmit our manuscript. We sincerely express our gratitude to editor and reviewers for the valuable comments. The editor’s and reviewers’ comments are laid out below in black text and specific concerns have been numbered. The revision has been finished, and the changes, additions and replies to reviewers' comments are listed below. All the changes, including text, equations and figures (captions), are also highlighted point by point using red text in the revised manuscript.Symbols and pictures are not shown in full in the website, please view the full content in Author's Notes File.
Once again, we really appreciate the works of editors and reviewers. These comments and suggestions have greatly helped to improve the manuscript quality. Please let us know if further modifications are required. Your reply will be highly appreciated. Sincerely thanks.
-------------------------------------------------------------------------------------------------------
-------------------------------------------------------------------------------------------------------
Comments from Reviewer #3:
-------------------------------------------------------------------------------------------------------
² Comment 1: This work presented the tested results and analysis on bending and angle effect on combustion behavior of U-shaped cables. Overall, this work is academically sound. A minor issue to address would be as this is an experimental study, please include uncertainties for the measured and deduced parameters.
- Response: Sincerely thanks to the reviewer’s timely reminder. Each set of experimental conditions was repeated at least three times. It should be pointed out that the experimental data naturally includes a certain degree of random error, which will inevitably be also involved in the subsequent analysis and empirical model correlation. Hence, the current data errors need to be evaluated firstly. In order to ensure the accuracy of the critical velocity in this study, we use the Eq. (1) to estimate the experimental data error.
,
(1)
where the Xi(measured) represents the observation in a single sample experiment or the mean of a series of n observations in a multiple sample experiment; σ is the standard deviation of measurements. The δXi represents 2σ for a single sample test (i.e. δXi=2σ ). The final estimation results imply that the maximum relative uncertainty of temperature data is not more than 12.56 % under the 95% confidence. Similarly, the uncertainty of mass loss data, flame spread rate and fame parameters are evaluated to be no more than 9.28%, 11.06%, and 14.85% respectively.
Once again, sincerely thanks to reviewer for the constructive comments. We have also added corresponding explanations in the revised manuscript. Thanks again, sincerely.
The specific changes/additions to manuscript are outlined below in red text:
It should be noted that the experimental data naturally includes a certain degree of random error, which will inevitably be also involved in the subsequent analysis and empirical model correlation. Hence, the current data errors need to be evaluated firstly. The uncertainty of experimental data can be estimated by
, . Where the represents the mean value of n observations in multiple sample experiment; σ is the standard deviation. The δXi represents 2σ for a single sample test (i.e. δXi=2σ). The final estimation results imply that the maximum relative uncertainty of temperature data is not more than 12.56 % under the 95% confidence. Similarly, the uncertainty of mass loss data, flame spread rate and fame parameters are evaluated to be no more than 9.28%, 11.06%, and 14.85% respectively.
² Comment 2: Minor language issues such as a few misspellings in the manuscript should be fixed..
- Response: Sincerely thanks to the reviewer’s timely reminder. Words in the paper have been carefully checked for spelling and grammar.
The specific changes/additions to manuscript are outlined below in red text:
(Page 10, line 273-275 ,of revised manuscript)
For the bending angle, it can be seen that the FSR increases with the bending angle. The FSR reaches the highest in UBD 90°, close to 6.81cm/min, which is four times high-er than that in NBM.
(Page 10, line 278-280 ,of revised manuscript)
Conversely, when U-shaped cables in DBM are heated, the melt would drip directly onto the floor along the cables surface, reducing diminishes the heating effect on the unburned region of the cables.
(Page 14, line 369-371 ,of revised manuscript)
Finally, it should be pointed that the current experimental study investigated the effect of different forms and angles on the combustion behaviour of U-shaped cables under middle ignition.
-------------------------------------------------------------------------------------------------------
Once again, we deeply appreciate the reviewers' recognition and precious comments, as well as the editor's patient response. These constructive suggestions are very important for our current work. Sincerely hope that with reviewers' and editor's guidance, we could improve the quality of our manuscript to the greatest extent and finally reach the published level in Fire.
Once again, sincerely thanks.

Round 2
Reviewer 1 Report
The author has addressed the reviewer's concerns and therefore accepts.
Reviewer 2 Report
The authors have properly addressed all my comments. The manuscript could be now considered for publication in the present form.